# Scaffold-Free Bone Regeneration Through Collaboration Between Type IV Collagen and FBXL14

**DOI:** 10.3390/jcm14207160

**Published:** 2025-10-11

**Authors:** Mari Akiyama

**Affiliations:** Department of Biomaterials, Osaka Dental University, 8-1, Kuzuhahanazono-cho, Hirakata, Osaka 573-1121, Japan; mari@cc.osaka-dent.ac.jp; Tel.: +81-72-864-3056

**Keywords:** type IV collagen, FBXL14, periosteum-derived cells, bone regeneration, scaffold-free

## Abstract

**Background**: The periosteum and periosteum-derived cells have attracted considerable attention for their potential use in clinical applications for treating bone defects. Bovine periosteum-derived cells have been investigated because of their capability for scaffold-free bone regeneration. Previous mass spectrometry (MS) and immunohistochemistry studies have shown the presence of F-box/leucine-rich repeat protein 14 (FBXL14) in bovine periosteum and periosteum-derived cells. Recently, studies using ESI-Q-Orbitrap MS suggested the presence of type IV collagen in the periosteum. The aim of the present study was to clarify the relationship between type IV collagen and FBXL14 in the formation of periosteum-derived cells. **Methods**: Bovine periosteum-derived cells were obtained from Japanese Black Cattle’s legs in Medium 199 with ascorbic acid and 10% fetal bovine serum. Immunohistochemistry for type IV collagen and FBXL14 was performed using bovine bone with periosteum and periosteum alone for explant culture. **Results**: Both type IV collagen and FBXL14 were expressed in Volkmann’s canals and the Haversian canals in bone and periosteum. After 5 weeks, type IV collagen and FBXL14 surrounded crystals containing osteocalcin and had formed periosteum-derived cells. Von Kossa staining and immunostaining of osteocalcin revealed that the crystals contained calcified substances and osteocalcin. **Conclusions**: Clinically, understanding osteocalcin-interacting proteins will help promote bone regeneration. Interactions between type IV collagen and FBXL14 may contribute to scaffold-free bone regeneration.

## 1. Introduction

The periosteum and periosteum-derived cells (PDCs) have attracted attention for clinical use in bone regeneration due to their osteogenic ability [1,2,3]. For tissue engineering, the periosteum and PDCs are often used with artificial materials [4,5]. There are two methods of culturing PDCs: collagenase digestion [1,6] and explant culture methods [7]. Because periosteum contains osteogenic stem cells [8], PDCs are periosteum-derived stem cells identified using stem cell markers [2]. In clinical use, PDCs are used for guided bone regeneration [3] with artificial synthetic scaffolds. However, these scaffolds have problems of rejection, shrinkage, or, quite the opposite, remain for a long time. Therefore, biomimetic scaffold-free techniques have been developed [9,10]. Bovine PDCs in the absence of any artificial material can form a multilayer structure made from extracellular matrix (ECM) in vitro (Figure 1), and scaffold-free PDCs resulted in bone regeneration in vivo when grafted into the backs of nude mice [11,12,13]. The molecular signaling of the periosteum involved in bone healing has also been investigated [1,14,15,16].

To clarify the osteogenic mechanism of protein interactions, we previously performed mass spectrometry (MS) and immunohistochemistry [17] and found that F-box/leucine-rich repeat protein 14 (FBXL14) supported the formation of multilayered PDCs [13,18]. FBXL14, an F-box protein, regulates the epithelial-to-mesenchymal transition with target proteins during development [19] and the progression of atherosclerotic plaque via the NRF2 signal pathway through ubiquitination of bispecific phosphatase-6 [20]. Osteocalcin is a vitamin K-dependent carboxylated protein that is related to osteoporosis [21,22,23,24]. Undercarboxylated osteocalcin is also related to whole-body metabolism as a hormone [25]. In 2018, Akiyama [26] reported that fibers of F-box and WD-40 domain-containing protein 2 (FBXW2) were coated with osteocalcin in the periosteum (Appendix A). In the periosteum, osteocalcin was assembled into crystal form without FBXW2 fibers [26]. These osteocalcin-containing crystals were identified by using double immunostaining methods in previous studies [26,27]. Electrospray ionization (ESI)-Q-Orbitrap MS suggested the presence of a basement membrane component, type IV collagen, with over 890 proteins in the supernatant of PDCs [28] (Appendix A), whereas electrospray ionization quadrupole time-of-flight (ESI-Q-TOF) MS/MS analysis was unable to detect type IV collagen [17]. Type IV collagen plays an important role in peripheral nerve myelin in regulating Schwann cells [29,30,31] and capillaries as a permeability selective barrier [32], and mutations in type IV collagen-related genes can cause disorders such as Alport syndrome [33,34,35]. Not only type IV collagen as the capillary basement membrane, but FBXL14 is involved in the capillaries of the periosteum [18] based on their presence in supernatants of periosteum using ESI-Q-TOF MS/MS analysis (Appendix A). The aim of the present study was to compare and clarify the relationship between type IV collagen and FBXL14 within the ECM of PDCs for bone regeneration.

## 2. Materials and Methods

Bovine skin, bone, and periosteum were obtained from the legs of 30-month-old Japanese Black Cattle (female and steer, Kobe Chuo Chikusan, Kobe, Japan) within 24 h after death for beef product, according to the Osaka Dental University Regulations on Animal Care and Use (Approval No. 25-02002). Six bovine legs were chosen randomly by the staff of a slaughterhouse; the researcher could not enter the slaughterhouse. This study did not involve living animals. All tissues were fixed with 4% paraformaldehyde. Bone with periosteum from two legs was cut with an Er:Yag Laser Device Erwin Adverl MEY-1 (pulse repetition; 25 pps, energy/pulse; 70 mJ, Morita Corp., Osaka, Japan), and bone was broken with a hammer. Bone and periosteum were demineralized with 10% ethylenediaminetetraacetic-2Na for 8 days at room temperature after fixation for sectioning. The periosteum from six bovine legs was removed with aseptic methods and used for explant culture, as previously described [18]. Bovine legs were disinfected with 10% povidone-iodine (iNova Pharmaceuticals Japan, Tokyo, Japan) and 10% sodium hypochlorite solution (Antiformin, FIJIFILM Wako Pure Chemical Corporation, Osaka, Japan). Approximately 5 mm^2^ periosteal pieces were cut using a disposable scalpel (No. 10, Feather Safety Razor Co., Ltd., Osaka, Japan) and put in a 100 mm-diameter culture dish (353003, FALCON, Corning Incorporated-Life Sciences, Durham, NC, USA) with 5 mg/mL ascorbic acid (A0276-25G Sigma-Aldrich-Merck Japan, Tokyo, Japan), penicillin/streptomycin (168-23191; FUJIFILM Wako Pure Chemical Corporation, Osaka, Japan), and 10% fetal bovine serum (Biosera, Cholet, France) in Medium 199 at 37 °C and 5% CO_2_ in an incubator and cultured for 5 weeks. The culture medium was changed once a week. The skin was prepared as a positive control for the basement membrane. All tissue samples were embedded in paraffin blocks and sectioned at a thickness of 2 μm for immunohistochemistry and von Kossa staining.

In accordance with datasets from ESI-Q-Orbitrap MS (JPST003206, JPST003258) in jPOSTrepo (Japan ProteOme STandard Repository), the presence of type IV collagen was suggested [28]. The activity of antigens in sections was retrieved using Proteinase K (Dako Cytomation, Glostrup, Denmark) for 10 min at room temperature. Anti-collagen Type IV (rabbit) antibody-600-401-1065 (1:500, 1 h; Rockland Immunochemicals, Inc., Rockland County, NY, USA) and anti-FBXL14 (rabbit) antibody (1:500 4 h; #SAB2103691; Sigma-Aldrich, Saint Louis, MO, USA) were used as the primary antibodies. Anti-EXOSC9 (1:500, 4 h; H-300; #sc-135118; Santa Cruz Biotechnology, Inc., Santa Cruz, CA, USA) (rabbit) antibody was used as the negative control. Alkaline phosphatase-conjugated goat anti-rabbit IgG(H+L) (1:200, 1 h; Proteintech Group, Inc., Rosemont, IL, USA) was used as the secondary antibody. All sections were visualized using PermaRed/AP (K049; Diagnostic BioSystems, Pleasanton, CA, USA). Images were obtained using a BZ-X810 microscope (Keyence Japan, Osaka, Japan) and X800 Viewer (Keyence). X800 Analyzer software (Keyence) was used to insert scale bars and increase the contrast. All images are preserved at the repository site GakuNin RDM (https://rdm.nii.ac.jp/)

To clarify the crystal components in the periosteum after 5 weeks, immunohistochemistry using antibody for osteocalcin (1:500, overnight; code no. M042, clone no. OCG2; Takara Bio Inc., Kusatsu, Japan), alkaline phosphatase-conjugated goat anti-mouse IgG(H+L) (1:200, 1 h; Proteintech Group), and PermaBlue/AP (K058; Diagnostic BioSystems) were used. From 2015 to 2017, double immunostaining of osteocalcin and other proteins was performed [26]. In previous studies, antibody for osteocalcin (1:500, 4 h; code no. M042, clone no. OCG2; Takara Bio Inc.) was used. N-Histofine Simple Stain AP (M) (#414241, Nichirei Biosciences Inc., Tokyo, Japan) was used as a secondary antibody. Rabbit uveal autoantigen with coiled-coil domains and ankyrin repeats (UACA) antibody (#bc-6308R; Bioss Inc., Woburn, MA, USA), N-Histofine Simple Stain AP (R) (#414251, Nichirei Biosciences Inc.), and PermaRed/AP were used for double immunostaining for UACA.

## 3. Results

### 3.1. Skin

Immunohistochemistry showed that type IV collagen was expressed strongly, not only in the basement membrane but also in microvessels, dermal papillae, and sweat glands (Figure 2, left). In contrast, FBXL14 was expressed strongly in sebaceous glands and the outer root sheath, but weakly in microvessels (Figure 2, right).

### 3.2. Bone and Periosteum

Type IV collagen and FBXL14 were expressed in the periosteum (Figure 3). In bone, type IV collagen and FBXL14 were expressed in Volkmann’s and Haversian canals. Expression of FBXL14 in Volkmann’s canals containing erythrocytes indicated the presence of FBXL14 in the capillaries (Figure 3, right, middle). However, in comparison with FBXL14, type IV collagen was expressed more extensively and strongly in capillaries because it was more abundant (Figure 3, middle, bottom).

### 3.3. Periosteum Explant Culture on Day 0

Figure 4 and Figure 5 show a comparison of type IV collagen and FBXL14 expressions in the periosteum on day 0. Figure 4 shows low-magnification images of the periosteum. Type IV collagen was expressed strongly in capillaries and widely throughout the periosteum. FBXL14 was mainly expressed in a single layer of the periosteum. Periosteum was divided into two layers, the cambium layer and the fibrous layer. The layer that strongly expressed type IV collagen and FBXL14 was the cambium layer, near bone (Figure 3 and Figure 4). Figure 5 shows low- and high-magnification images of the periosteum. In Figure 5, a fine mesh of type IV collagen surrounded the capillary wall, whereas FBXL14 was expressed partially within the capillary wall.

### 3.4. Periosteum Explant Culture at 5 Weeks

After 5 weeks, type IV collagen and FBXL14 formed a matrix in which the PDCs were supported. Type IV collagen surrounded crystals, and FBXL14-positive cells were assembled around these crystals (Figure 6, Figure 7 and Figure 8, red arrows). Type IV collagen and the assembled FBXL14-positive cells formed a membrane around crystals and the complex of PDCs, and the ECM appeared to flake off from the periosteum (Figure 6, Figure 7 and Figure 8, red arrows). As shown in the bottom images of Figure 8, part of the periosteum within the ECM had shifted. Figure 9 shows a comparison of von Kossa staining and immunostaining of osteocalcin. The crystals are calcified and contain osteocalcin. Figure 10 shows a fibrous structure in the periosteum. Type IV collagen surrounded this fibrous structure and was expressed strongly, whereas FBXL14 was expressed in the fibrous structure inside the type IV collagen membrane. Immunohistochemistry indicated the presence of a translucent membrane around FBXL14 (Figure 10, bottom, right).

Figure 11 shows negative controls using anti-EXOSC9 antibody. The skin and periosteum (on day 0 and at 5 weeks) did not exhibit any anti-EXOSC9 antibody reaction.

## 4. Discussion

The present study first aimed to clarify whether FBXL14 is a basement membrane component like type IV collagen, using immunohistochemistry for type IV collagen as a positive control. Skin was used for basement membrane detection. Type IV collagen is present in the basement membrane of capillaries [35] and hair follicles [36]. The expression patterns of type IV collagen and FBXL14 differ in skin (Figure 2). The periosteum contains sensory nerves [37], and type IV collagen is an important component in the myelin of peripheral nerves [29,30,31]. As shown in Figure 10, type IV collagen and FBXL14 were clearly expressed in different regions of the fibrous structure in the periosteum. S-100 protein, a nerve marker, has also been shown to be present in different regions of nerve tissue from type IV collagen [38]. However, without nerve markers, such as S-100 protein, it cannot be determined, but the fibrous structure in Figure 10 thus could be a sensory nerve.

In bone, capillaries, and the periosteum, type IV collagen and FBXL14 were expressed in similar regions, although FBXL14 expression was weaker. There is an inner cambium layer between bone and the periosteum [39]. As shown in Figure 3, type IV collagen and FBXL14 were expressed strongly in the cambium layer near bone. The periosteum is divided into two layers: the cambium layer (with bone) and the fibrous layer (detached from bone) [1]. The cambium layer could be the region in which type IV collagen and FBXL14 are strongly expressed in Figure 4, which shows the periosteum without bone. As shown in Figure 5, FBXL14 expression in capillaries was much weaker than that of type IV collagen, suggesting that type IV collagen was located outside areas in which FBXL14 was expressed. In this study, two rabbit primary antibodies, anti-collagen type IV antibody (1:500, 1 h) and anti-FBXL14 antibody (1:500 4 h), were used. Anti-EXOSC9 (1:500 4 h) antibody was used for negative controls because anti-EXOSC9 antibody is also a rabbit primary antibody and proves the absence of non-specific staining.

Type IV collagen and FBXL14 were involved in the formation of PDCs over the course of the 5-week explant culture in the present study. First, type IV collagen surrounded crystals. Next, FBXL14 formed a multilayered complex of ECM and PDCs with type IV collagen inside the periosteum. Finally, PDCs appeared to flake off outside the periosteum. In this study, crystal structures were observed in four of six cows (Table 1). In cow 2, no crystal structure was observed because the sections were thin, and these thin sections did not include crystal structures. In cow 6, PDCs were not observed. These differences were observed because the bovine legs were chosen randomly without any standard conditions (such as temperature) in the slaughterhouse. In this case, periosteal cells inside tissue became weak. According to a report by Akiyama [26], the crystal in the periosteum contains osteocalcin (Figure 9, right), which regulates bone mineralization [40]. Figure 9 and Appendix A show the crystal is possibly a complex of calcium salt and osteocalcin. Osteocalcin binds to calcium phosphate [41]. During the 5 weeks of explant culture, osteocalcin likely bound to calcified crystals in the periosteum. In a previous study, treatment with anti-osteocalcin antibody was for 4 h [26], whereas in the present study, it was overnight, because the secondary antibody and the lot number of the primary antibody were different. The results of the present study suggest that osteocalcin is also related to the synthesis of the ECM. In the periosteum, Wnt signaling [14], Dickkopf-related protein 3 [15], the E3 ubiquitin ligase, and the Cbl–phosphatidylinositol-3 kinase interaction [16] are important for bone regeneration. Type I collagen and osteocalcin, a non-collagenous protein, reportedly interact during bone metabolism [42]. Types I and III collagens were abundant in the ECM of PDCs [17]. The supporting results of the previous and present studies indicate that collagen types I, III, and IV and FBXL14 interact during the formation of multilayered PDCs (Figure 12).

Previous studies using ESI-Q-TOF MS/MS [17] and ESI-Q-Orbitrap MS [25] investigated the supernatants of PDCs. ESI-Q-TOF MS/MS was able to detect a few proteins associated with type I and III collagens, but not type IV collagen [17]. ESI-Q-Orbitrap MS detected over 890 proteins associated with type IV collagen [28]. However, the numerous proteins identified by ESI-Q-Orbitrap MS made it difficult to understand which proteins actually affect bone regeneration. The combination of immunohistochemistry and MS methods has revealed interactions between collagen and non-collagenous F-box proteins, but the sensitivity and precision of the antibodies used for the immunohistochemistry method were limited. In the future, more detailed investigations using mass spectrometry imaging will be needed to further clarify the interactions between collagen and non-collagenous proteins.

## 5. Conclusions

FBXL14 cannot be considered a basement membrane protein. However, type IV collagen and FBXL14 interact with osteocalcin during the formation of PDCs. Clinically, understanding osteocalcin-interacting proteins will help clarify osteoporosis. Biomimetics using native ECM including collagen types IV and FBXL14 may be useful for novel scaffold-free methods in tissue engineering.

## Figures and Tables

**Figure 1 jcm-14-07160-f001:**
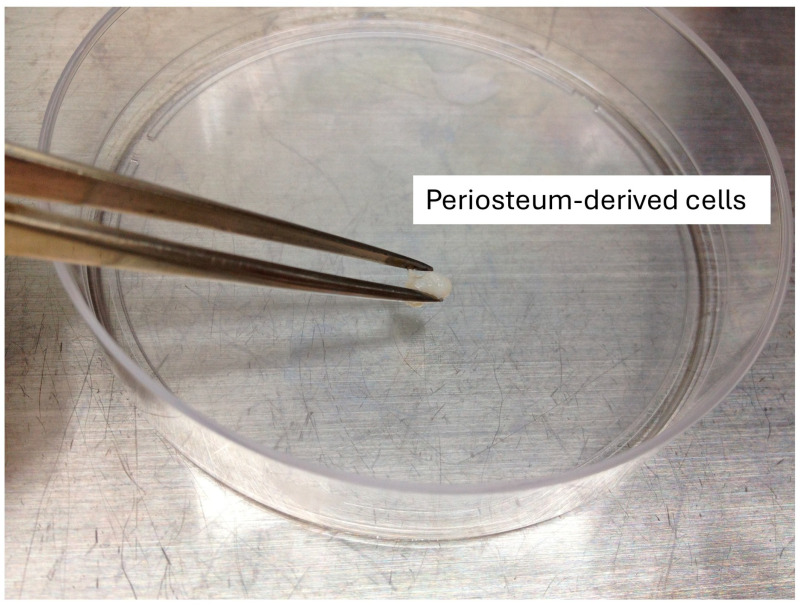
Periosteum-derived cells for scaffold-free bone regeneration after 5 weeks of explant culture.

**Figure 2 jcm-14-07160-f002:**
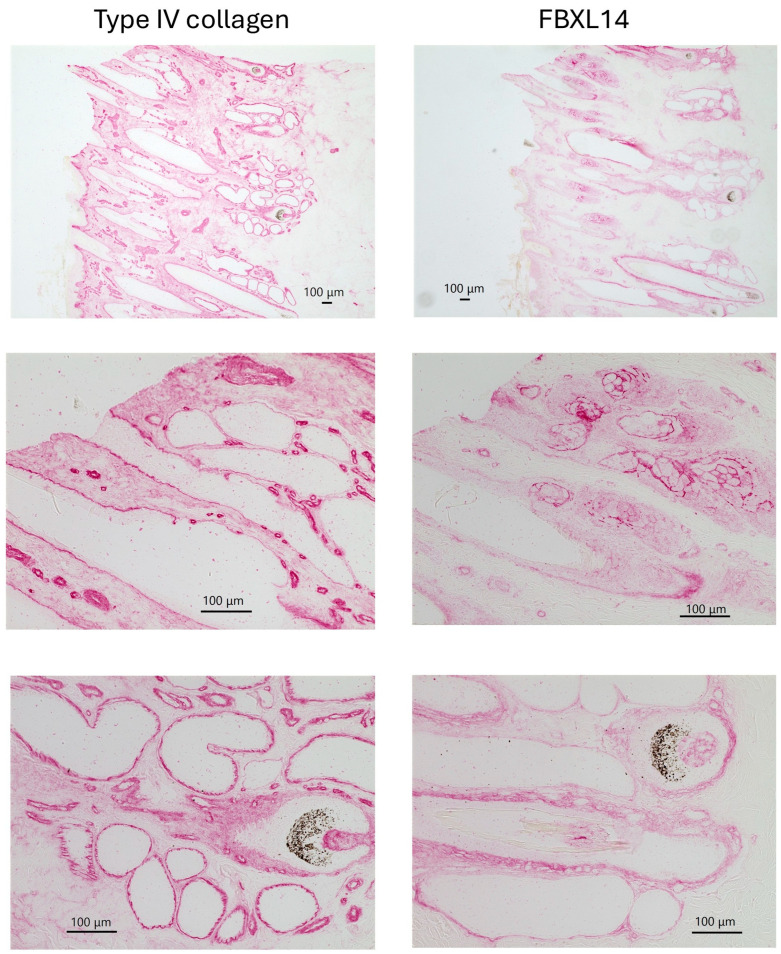
Comparison of immunohistochemistry results for type IV collagen and FBXL14 in bovine skin.

**Figure 3 jcm-14-07160-f003:**
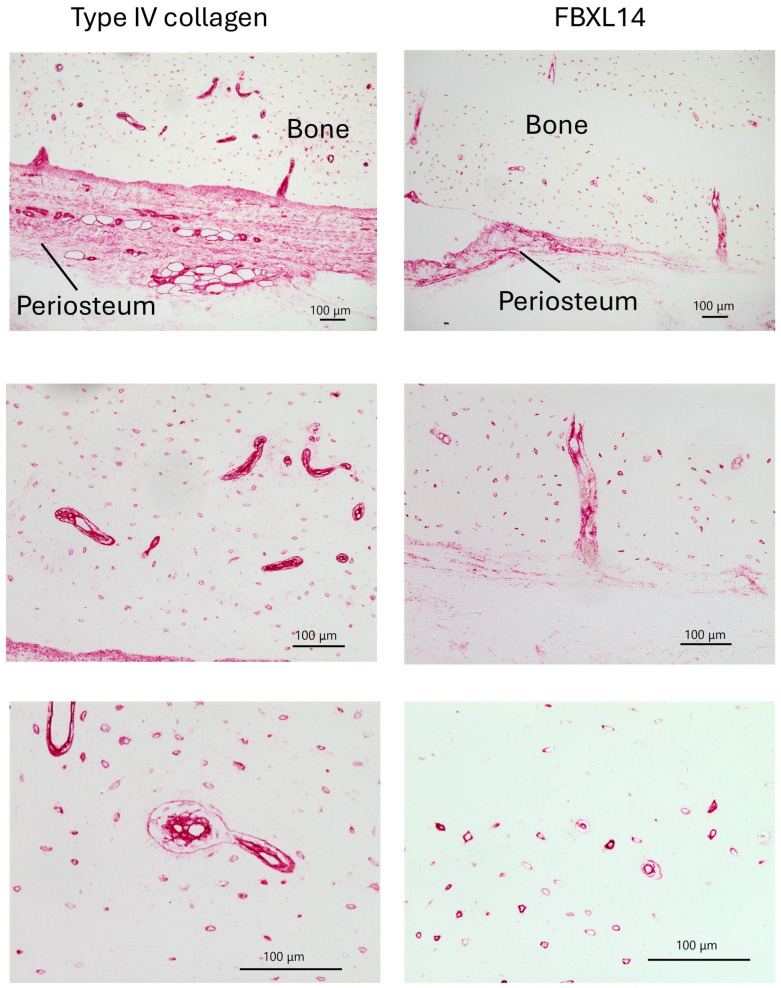
Immunohistochemistry results for type IV collagen and FBXL14 in bovine bone and periosteum. Type IV collagen and FBXL14 are expressed in the periosteum and in Volkmann’s and Haversian canals in bone. Type IV collagen was more prevalent than FBXL14 in the bone and periosteum.

**Figure 4 jcm-14-07160-f004:**
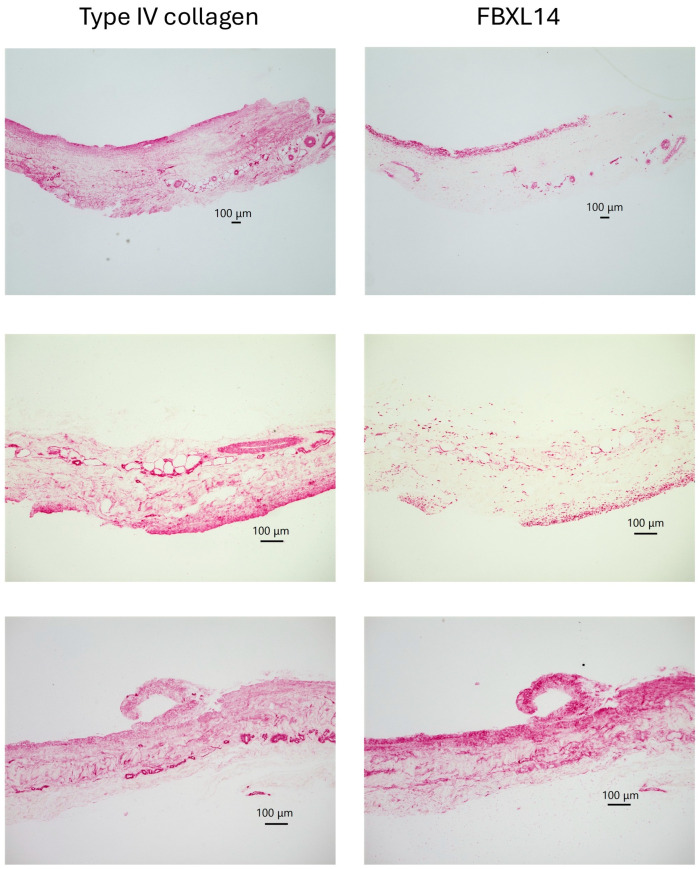
Low-magnification images of the periosteum without bone on day 0.

**Figure 5 jcm-14-07160-f005:**
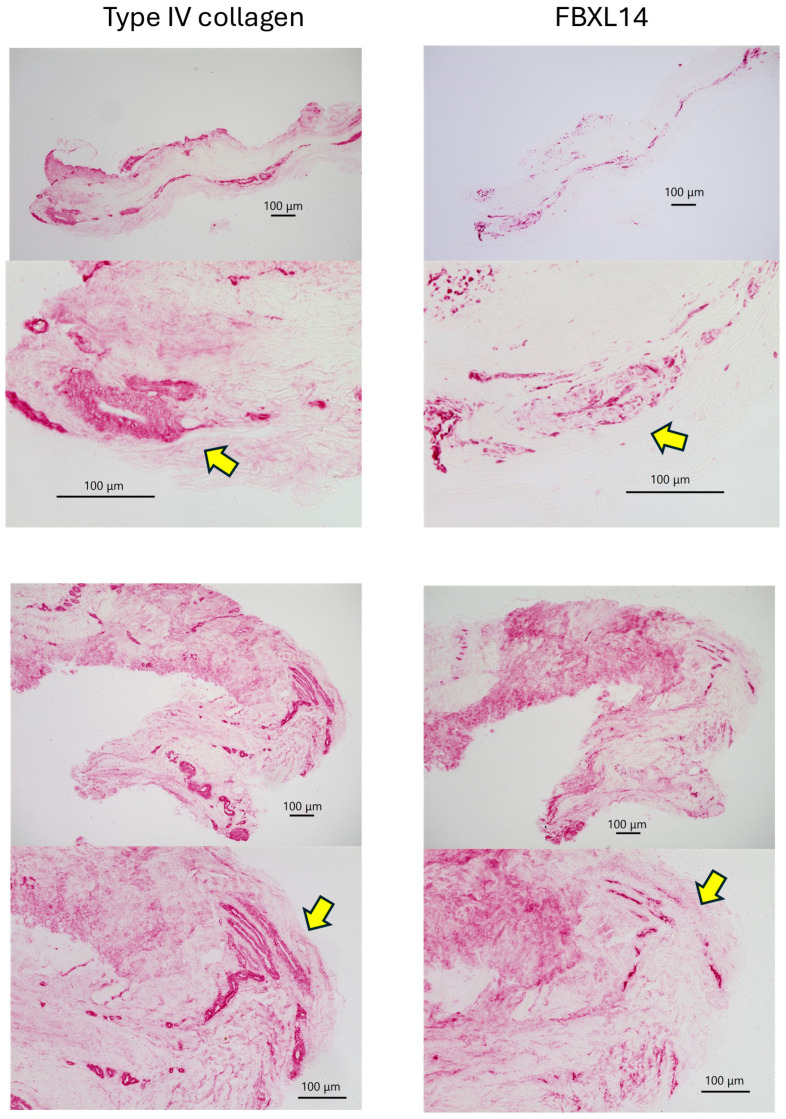
Comparison of low- and high-magnification images of the periosteum on day 0. A fine mesh of type IV collagen and diffuse expression of FBXL14 are observed in capillaries (yellow arrows).

**Figure 6 jcm-14-07160-f006:**
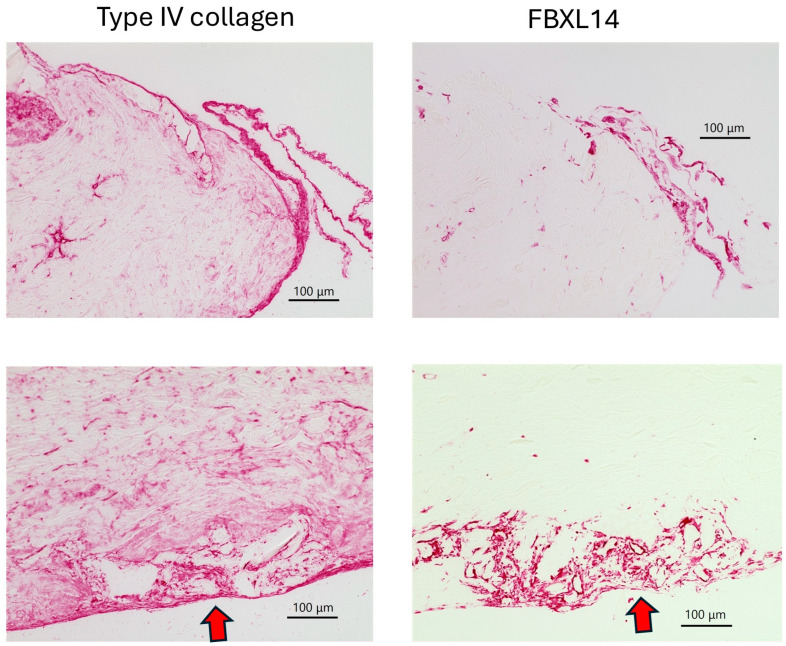
The periosteum after 5 weeks of explant culture. The crystals likely contain osteocalcin (red arrows), and periosteum-derived cells are present.

**Figure 7 jcm-14-07160-f007:**
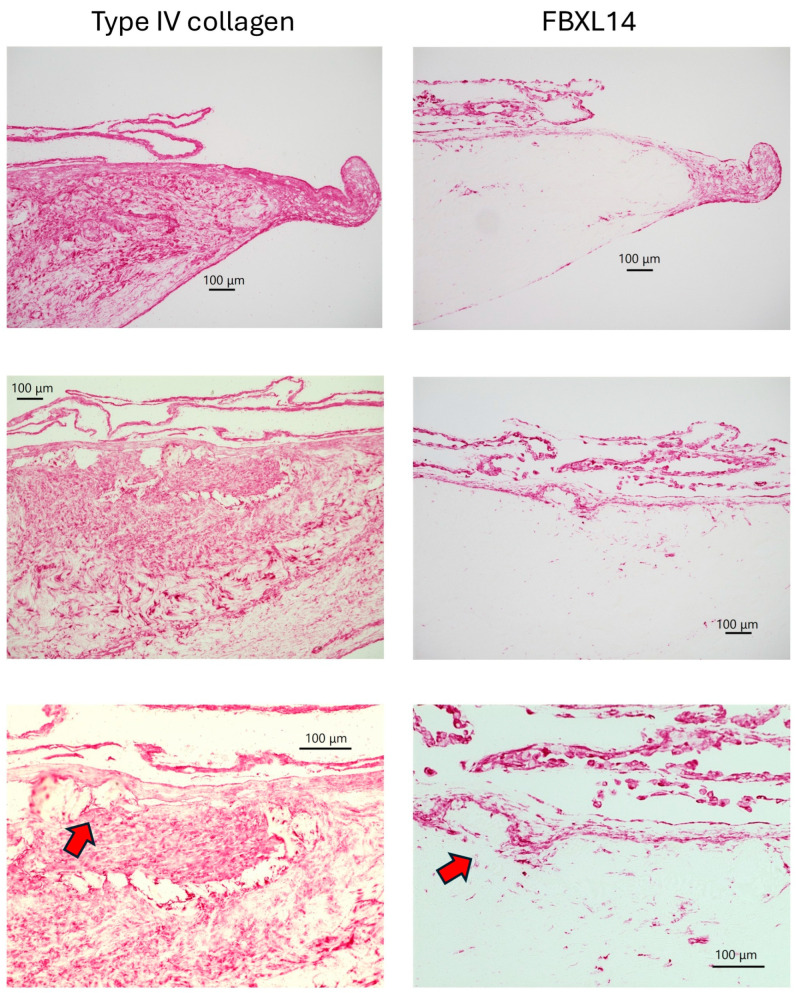
Additional images of the periosteum after 5 weeks of explant culture show the presence of crystals (red arrows) and periosteum-derived cells.

**Figure 8 jcm-14-07160-f008:**
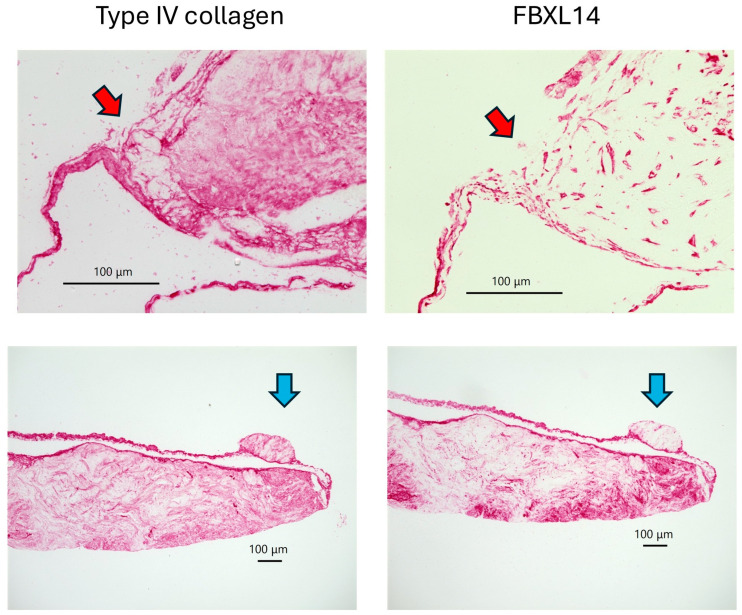
Other images of the periosteum after 5 weeks of explant culture also show the presence of crystals (red arrows) and periosteum-derived cells. Part of the periosteum within the ECM (blue arrows) has shifted.

**Figure 9 jcm-14-07160-f009:**
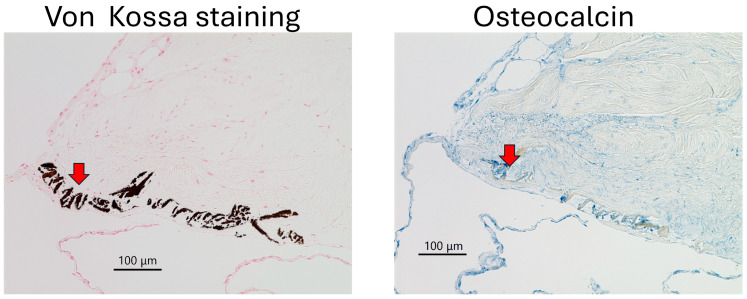
Von Kossa staining and immunostaining of osteocalcin (blue) of the periosteum after 5 weeks of explant culture also showed the presence of crystals (red arrows).

**Figure 10 jcm-14-07160-f010:**
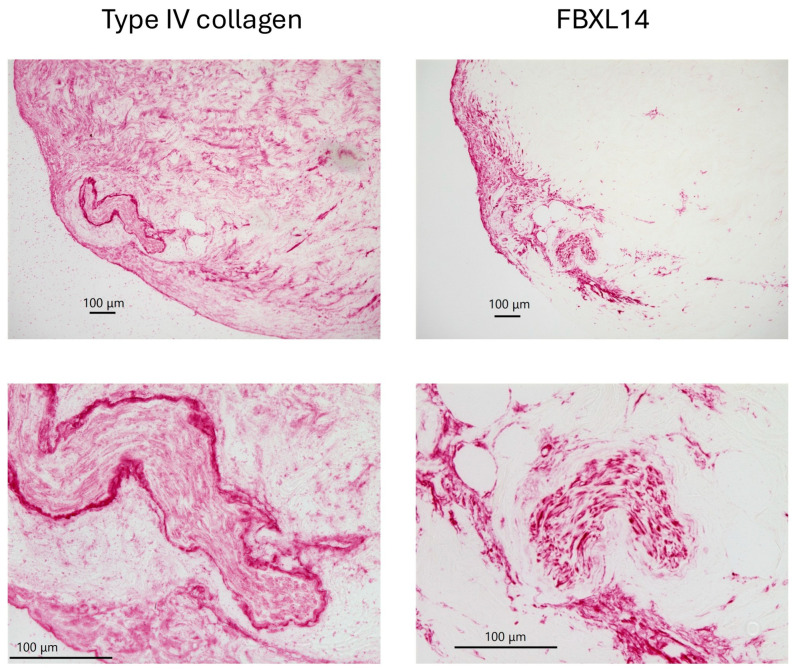
Comparison of type IV collagen and FBXL14 expression in the fibrous structure of the periosteum after 5 weeks of explant culture.

**Figure 11 jcm-14-07160-f011:**
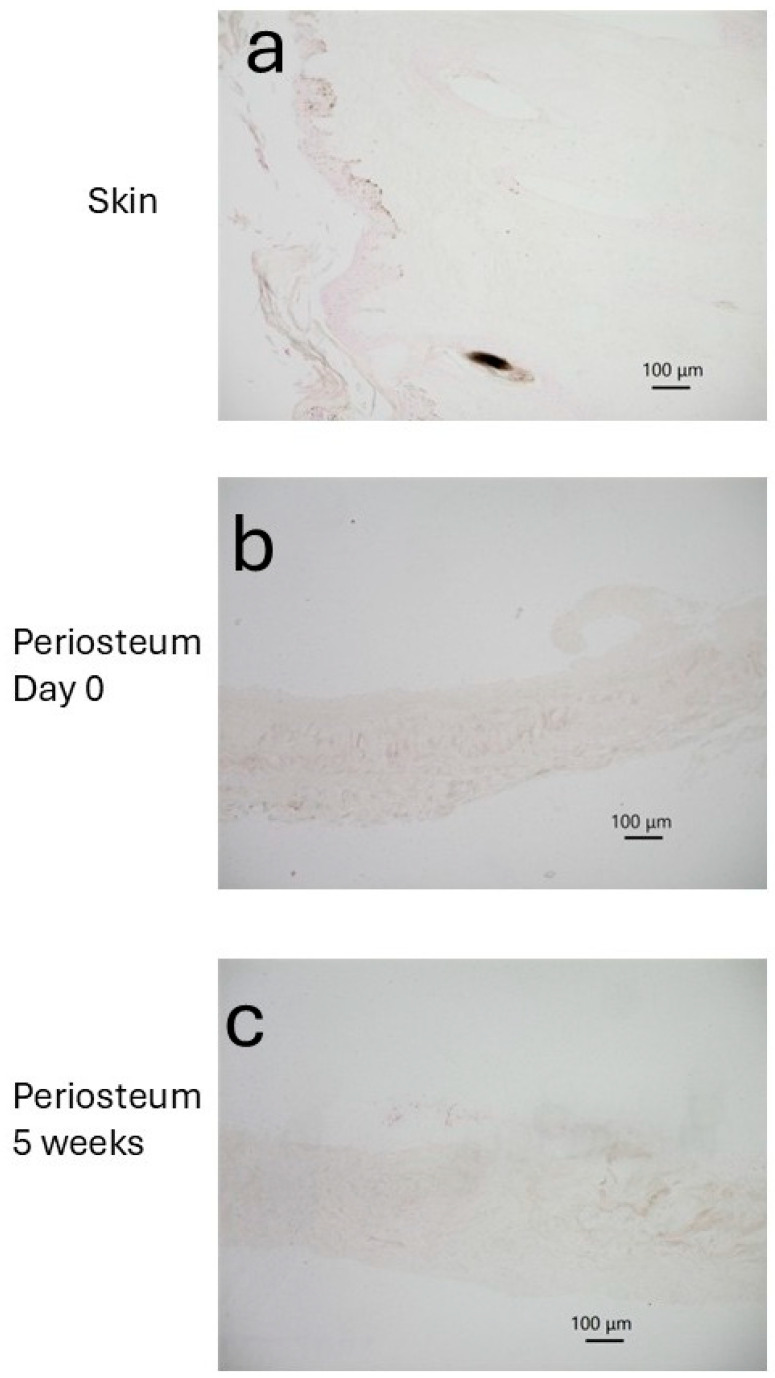
Negative controls: (**a**) Skin; (**b**) Periosteum on day 0; (**c**) Periosteum at 5 weeks.

**Figure 12 jcm-14-07160-f012:**
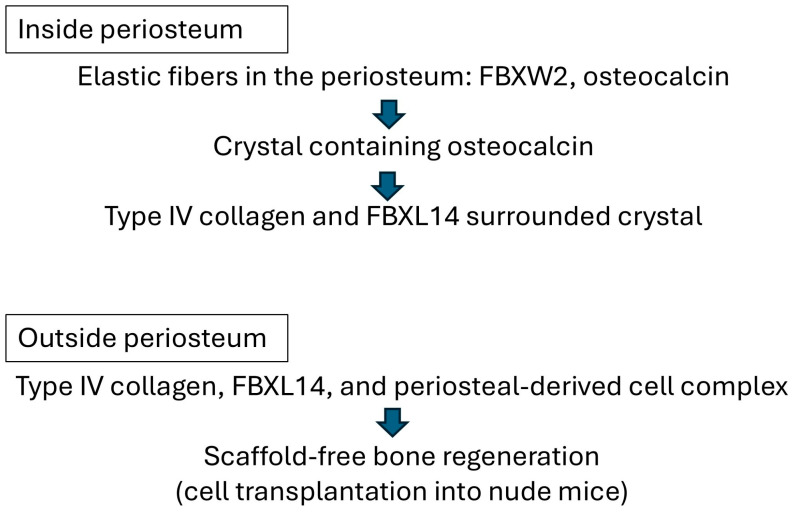
Hypothetical process of scaffold-free bone regeneration from the previous and present studies.

**Table 1 jcm-14-07160-t001:** Results from six cows.

Cow 1.	PDCs with Type IV collagen and FBxL14, crystal
Cow 2.	PDCs with Type IV collagen and FBxL14
Cow 3.	PDCs with Type IV collagen and FBxL14, crystal
Cow 4.	PDCs with Type IV collagen and FBxL14, crystal
Cow 5.	PDCs with Type IV collagen and FBxL14, crystal
Cow 6.	PDCs not observed.

## Data Availability

Datasets from ESI-Q-Orbitrap MS (JPST003206, JPST003258) were deposited in jPOSTrepo (Japan ProteOme STandard Repository).

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
