# Peer review of "Scaffold-Free Bone Regeneration Through Collaboration Between Type IV Collagen and FBXL14"

_jcm, 2025, doi:10.3390/jcm14207160_

Round 1

Reviewer 1 Report

Comments and Suggestions for Authors
  1. The periosteum and periosteal-derived cells (PDCs) have attracted attention for clinical use in bone regeneration due to their osteogenic ability [1-10].  There are too many references cited here in introduction, only 2-3 representative literature with high impact factors should be cited, for example: 

    10.1016/j.jot.2022.01.002.       10.1016/j.actbio.2024.12.058.

  2. There is no relevant description of Table S1 in the supplementary materials in the main text. What does this data represent that needs to be introduced.
  3. Why test the skin?

Author Response

  1. The periosteum and periosteal-derived cells (PDCs) have attracted attention for clinical use in bone regeneration due to their osteogenic ability [1-10].  There are too many references cited here in introduction, only 2-3 representative literature with high impact factors should be cited, for example: 

10.1016/j.jot.2022.01.002.       10.1016/j.actbio.2024.12.058.

Thank you for your insightful comment. The alterations to the manuscript are shown below and are highlighted in red within the revised manuscript.

Line 32-33:

The periosteum and periosteum-derived cells (PDCs) have attracted attention for clinical use in bone regeneration due to their osteogenic ability [1-3].

  1. Lin, Z.; Fateh, A.; Salem, D.M.; Intini, G. Periosteum: biology and applications in craniofacial bone regeneration. J Dent Res 2014, 93, 109-116, doi:10.1177/0022034513506445.
  2. Zhang, X.; Deng, C.; Qi, S. Periosteum Containing Implicit Stem Cells: A Progressive Source of Inspiration for Bone Tissue Regeneration. Int J Mol Sci 2024, 25, doi:10.3390/ijms25042162.
  3. Zhang, W.; Wang, N.; Yang, M.; Sun, T.; Zhang, J.; Zhao, Y.; Huo, N.; Li, Z. Periosteum and development of the tissue-engineered periosteum for guided bone regeneration. J Orthop Translat 2022, 33, 41-54, doi:10.1016/j.jot.2022.01.002.
  1. There is no relevant description of Table S1 in the supplementary materials in the main text. What does this data represent that needs to be introduced.

Line 229-234: Table S1 was changed to Table 1.

In this study, crystal structures were observed in four of six cows (Table 1). In cow 2, no crystal structure was observed because the sections were thin, and these thin sections did not include crystal structures. In cow 6, PDCs were not observed. These differences were observed because the bovine legs were chosen randomly without any standard conditions (such as temperature) in the slaughterhouse. In this case, periosteal cells inside tissue became weak.

  1. Why test the skin?

Line201:

Skin was used for basement membrane detection.

Reviewer 2 Report

Comments and Suggestions for Authors

The major concern with this paper is that its underlying logic is flawed. The experimental design is overly simplistic and does not adequately support the authors’ main conclusion. The study only uses histology and explant experiments to show that type IV collagen and FBXL14 appear during the formation of periodontal ligament–derived cells. However, this merely demonstrates correlation rather than causation and does not substantiate the title’s claim: “Scaffold-Free Bone Regeneration via Type IV Collagen and FBXL14 Interaction.” To justify the use of “via,” a more rigorous approach would be required, such as selective gene knockout models (animal or at least in vitro cell systems), combined with rescue experiments, to provide preliminary validation of the proposed mechanism.

Additional issues:

  1. The title is not appropriate, as it does not accurately reflect the main content. Instead, it emphasizes a speculative deduction that is not sufficiently supported by the results.
  2. Can “the formation of periosteal-derived cells” truly be equated with bone regeneration? This conceptual leap is not justified in the manuscript.
  3. The definition of PDC is presented in a broad, vague, and generalized manner. Does the term refer to stem cells, or something else? Furthermore, this section cites an excessive number of references simultaneously, which undermines specificity.
  4. Lines 32–33: What is the clinical application of PDC? The introduction lacks sufficient background information to orient the reader.
  5. Line 34: The phrase “solve the deficiency of artificial material” is too vague and inaccurate as a way of introducing the concept. It requires clearer and more precise explanation.
  6. References 11 and 12 are very outdated and should be replaced with more recent and relevant literature.
  7. The figures are of very low quality, and the table included is more appropriate for a review article than for an original research paper.

Author Response

The major concern with this paper is that its underlying logic is flawed. The experimental design is overly simplistic and does not adequately support the authors’ main conclusion. The study only uses histology and explant experiments to show that type IV collagen and FBXL14 appear during the formation of periodontal ligament–derived cells. However, this merely demonstrates correlation rather than causation and does not substantiate the title’s claim: “Scaffold-Free Bone Regeneration via Type IV Collagen and FBXL14 Interaction.” To justify the use of “via,” a more rigorous approach would be required, such as selective gene knockout models (animal or at least in vitro cell systems), combined with rescue experiments, to provide preliminary validation of the proposed mechanism.

Additional issues:

  1. The title is not appropriate, as it does not accurately reflect the main content. Instead, it emphasizes a speculative deduction that is not sufficiently supported by the results.

Thank you for your insightful comment. Based on the reviewer’s suggestions, the title of the manuscript has been changed to the current title, “Scaffold-Free Bone Regeneration Through Collaboration between Type IV collagen and FBXL14”

  1. Can “the formation of periosteal-derived cells” truly be equated with bone regeneration? This conceptual leap is not justified in the manuscript.

The alterations to the manuscript are shown below and are highlighted in red within the revised manuscript.

Line 40-43:

Bovine PDCs in the absence of any artificial material can form a multilayer structure made from extracellular matrix (ECM) in vitro (Figure 1), and scaffold-free PDCs resulted in bone regeneration in vivo when grafted into the backs of nude mice [11-13].

  1. The definition of PDC is presented in a broad, vague, and generalized manner. Does the term refer to stem cells, or something else? Furthermore, this section cites an excessive number of references simultaneously, which undermines specificity.

Line 34-37:

There are two methods of culturing PDCs: collagenase digestion [1,6] and explant culture methods [7]. Because periosteum contains osteogenic stem cells [8], PDCs are periosteum-derived stem cells identified using stem cell markers [2].

  1. Lines 32–33: What is the clinical application of PDC? The introduction lacks sufficient background information to orient the reader.

Line 37-38:

In clinical use, PDCs are used for guided bone regeneration [3] with artificial synthetic scaffolds.

  1. Line 34: The phrase “solve the deficiency of artificial material” is too vague and inaccurate as a way of introducing the concept. It requires clearer and more precise explanation.

Line 38-39:

However, these scaffolds have problems of rejection, shrinkage, or, quite the opposite, remain for a long time.

  1. References 11 and 12 are very outdated and should be replaced with more recent and relevant literature.

Line 42-43: Reference [13] was added.

and scaffold-free PDCs resulted in bone regeneration in vivo when grafted into the backs of nude mice [11-13].

  1. Akiyama, M.; Nonomura, H.; Kamil, S.H.; Ignotz, R.A. Periosteal cell pellet culture system: a new technique for bone engineering. Cell Transplant 2006, 15, 521-532, doi:10.3727/000000006783981765.
  2. Akiyama, M.; Nakamura, M. Bone regeneration and neovascularization processes in a pellet culture system for periosteal cells. Cell Transplant 2009, 18, 443-452, doi:10.3727/096368909788809820.
  3. Akiyama, M. Characterization of the F-box Proteins FBXW2 and FBXL14 in the Initiation of Bone Regeneration in Transplants given to Nude Mice. Open Biomed Eng J 2018, 12, 75-89, doi:10.2174/1874120701812010075.

  1. The figures are of very low quality, and the table included is more appropriate for a review article than for an original research paper.

All images were increased the contrast. Table 1 was changed to Table S1. However, Table S1 was changed to Table 1.

Reviewer 3 Report

Comments and Suggestions for Authors

Dear Authors,

In this manuscript, authors investigated the interaction between type IV collagen and FBXL14 in the formation of periosteum-derived cells. They got bovine skin, bone and periosteum from Black Cattle, and type IV collagen and FBXL14 was performed using bovine bone with periosteum and periosteum alone for explant culture. Both type IV collagen and FBXL14 were expressed in bone and periosteum by IHC. Crystals which likely osteocalcin were covered by type IV collagen and FBXL14 and had formed periosteal-derived cells. They concluded that type IV collagen and FBXL14 will contribute to scaffold-free bone regeneration.

I have annotated the manuscript with several corrections, which I believe will improve the readability of the paper.

There are some concerns described below,

1, All image is not clear. Please use a higher resolution image.

2, In figure.5, authors described that crystals are likely osteocalcin, but it doesn’t show the evidence. It is important throughout the paper to confirm whether these crystals are osteocalcin.

3, Fig.9, A negative control always requires a corresponding experimental condition for comparison. However, it is currently unclear to the reader how this negative control relates to the preceding figures or data. It is important to clarify which specific experiment this negative control is intended to support. It should be described why you use the Anti-Exosc9 anti-body was used as the negative control.

Author Response

Dear Authors,

In this manuscript, authors investigated the interaction between type IV collagen and FBXL14 in the formation of periosteum-derived cells. They got bovine skin, bone and periosteum from Black Cattle, and type IV collagen and FBXL14 was performed using bovine bone with periosteum and periosteum alone for explant culture. Both type IV collagen and FBXL14 were expressed in bone and periosteum by IHC. Crystals which likely osteocalcin were covered by type IV collagen and FBXL14 and had formed periosteal-derived cells. They concluded that type IV collagen and FBXL14 will contribute to scaffold-free bone regeneration.

I have annotated the manuscript with several corrections, which I believe will improve the readability of the paper.

There are some concerns described below,

Thank you for your insightful comment. The alterations to the manuscript are shown below and are highlighted in red within the revised manuscript.

1, All image is not clear. Please use a higher resolution image.

All images were increased the contrast.

2, In figure.5, authors described that crystals are likely osteocalcin, but it doesn’t show the evidence. It is important throughout the paper to confirm whether these crystals are osteocalcin.

Figure 5 was changed to Figure 6.

New Figure 9 and Supplementary Figure S1 showed the presence of osteocalcin.

Line 56-57:

These osteocalcin-containing crystals were identified by using double immunostaining methods in previous studies [26,27].

Line 112-123:

To clarify the crystal components in the periosteum after 5 weeks, immunohistochemistry using antibody for osteocalcin (1:500, overnight; code no. M042, clone no. OCG2; Takara Bio Inc., Shiga, Japan), alkaline phosphatase-conjugated goat anti-mouse IgG(H+L) (1:200, 1 h; Proteintech Group), and PermaBlue/AP (K058; Diagnostic BioSystems) were used. From 2015 to 2017, double immunostaining of osteocalcin and other proteins was performed [26]. In previous studies, antibody for osteocalcin (1:500, 4h; code no. M042, clone no. OCG2; Takara Bio Inc.) was used. N-Histofine Simple Stain AP (M) (#414241, Nichirei Biosciences Inc.) was used as a secondary antibody. Rabbit uveal autoantigen with coiled-coil domains and ankyrin repeats (UACA) antibody (#bc-6308R; Bioss Inc., Woburn, MA, USA), N-Histofine Simple Stain AP (R) (#414251, Nichirei Biosciences Inc., Tokyo, Japan), and PermaRed/AP were used for double immunostaining for UACA.

3, Fig.9, A negative control always requires a corresponding experimental condition for comparison. However, it is currently unclear to the reader how this negative control relates to the preceding figures or data. It is important to clarify which specific experiment this negative control is intended to support. It should be described why you use the Anti-Exosc9 anti-body was used as the negative control.

Line 220-224:

In this study, two rabbit primary antibodies, anti-collagen type IV antibody (1:500, 1 h) and anti-FBXL14 antibody (1:500 4 h), were used. Anti-EXOSC9 (1:500 4 h) antibody was used for negative controls because anti-EXOSC9 antibody is also a rabbit primary antibody and proves the absence of non-specific staining.

Reviewer 4 Report

Comments and Suggestions for Authors

The study is relevant, but some adjustments could improve and complement the manuscript.

Considerations to the Authors:

1 - Introduction: Provide a better justification for the study, considering the gaps in the literature on the subject.

2 - Materials and Methods: Provide more detailed information on the methodology, considering the analyses performed, according to the sections detailed in the results.

3 - Discussion: Provide a more detailed discussion of the results in relation to the available literature, addressing issues such as the roles of type IV collagen and FBXL14 in tissues and biological processes, including osteogenesis and bone regeneration, considering the regulatory interactions between them and protein degradation.

Author Response

The study is relevant, but some adjustments could improve and complement the manuscript.

Thank you for your insightful comment. The alterations to the manuscript are highlighted in red within the revised manuscript.

Considerations to the Authors:

1 - Introduction: Provide a better justification for the study, considering the gaps in the literature on the subject.

Line 34-43, Line 56-57, New Figure1:

Explanation for the periosteum-derived cells and osteocalcin was added.

2 - Materials and Methods: Provide more detailed information on the methodology, considering the analyses performed, according to the sections detailed in the results.

Line 76-78:

Six bovine legs were chosen randomly by the staff of a slaughterhouse; the researcher could not enter the slaughterhouse. This study did not involve living animals.

Line 95-96:

and sectioned at a thickness of 2 μm for immunohistochemistry and von Kossa staining.

Line 112-123:

To clarify the crystal components in the periosteum after 5 weeks, immunohisto-chemistry using antibody for osteocalcin (1:500, overnight; code no. M042, clone no. OCG2; Takara Bio Inc., Shiga, Japan), alkaline phosphatase-conjugated goat anti-mouse IgG(H+L) (1:200, 1 h; Proteintech Group), and PermaBlue/AP (K058; Diagnostic BioSys-tems) were used. From 2015 to 2017, double immunostaining of osteocalcin and other pro-teins was performed [26]. In previous studies, antibody for osteocalcin (1:500, 4h; code no. M042, clone no. OCG2; Takara Bio Inc.) was used. N-Histofine Simple Stain AP (M) (#414241, Nichirei Biosciences Inc.) was used as a secondary antibody. Rabbit uveal au-toantigen with coiled-coil domains and ankyrin repeats (UACA) antibody (#bc-6308R; Bi-oss Inc., Woburn, MA, USA), N-Histofine Simple Stain AP (R) (#414251, Nichirei Biosci-ences Inc., Tokyo, Japan), and PermaRed/AP were used for double immunostaining for UACA.

3 - Discussion: Provide a more detailed discussion of the results in relation to the available literature, addressing issues such as the roles of type IV collagen and FBXL14 in tissues and biological processes, including osteogenesis and bone regeneration, considering the regulatory interactions between them and protein degradation.

New Figure 12 shows hypothetical process of scaffold-free bone regeneration.

Line 236-238:

Figure 9 and Supplementary Figure S1 show the crystal is possibly complex of calcium salt and osteocalcin. Osteocalcin binds to calcium phosphate [41]. For 5 weeks of explant culture, osteocalcin likely binds to calcified crystals in the periosteum.

Round 2

Reviewer 3 Report

Comments and Suggestions for Authors

Dear author,

I carefully reviewed your paper which was adequately improved.   The figures could show 

clearer contrast than before, and localization of oseocalcin was confirmed  with immunohistochemistry.

I strongly think that the contents of the paper become more powerful inclusion of additional results. I realized the potential value of the described data.

Best regards,

Reviewer 4 Report

Comments and Suggestions for Authors

The adjustments improved and complemented the manuscript,